# Comparison of Properties of 3D-Printed Mortar in Air vs. Underwater

**DOI:** 10.3390/ma14195888

**Published:** 2021-10-08

**Authors:** Seong-Jin Woo, Jun-Mo Yang, Hojae Lee, Hong-Kyu Kwon

**Affiliations:** 1Department of Civil Engineering, Keimyung University, 1095 Dalgubeol-daero, Dalseo-gu, Daegu 42601, Korea; woo.sj@kmu.kr; 2Korea Institute of Civil Engineering and Building Technology, Daehwa-dong, Goyang-si 10223, Korea; h.lee@kict.re.kr; 3Department of Industrial and Management Engineering, Namseoul University, 91 DaeHakro, Seonghwan-eup, Cheonan-si 31020, Korea; hongkyuk@nsu.ac.kr

**Keywords:** 3D concrete printing, 3D printing underwater, mortar printing, compressive strength, flexural tensile strength, interlayer bond strength, density

## Abstract

Research and technological advancements in 3D concrete printing (3DCP) have led to the idea of applying it to offshore construction. The effect of gravity is reduced underwater, which can have a positive effect on 3DCP. For basic verification of this idea, this study printed and additively manufactured specimens with the same mortar mixture in air and underwater and evaluated properties in the fresh state and the hardened state. The mechanical properties were evaluated using the specimens produced by direct casting to the mold and specimens produced by extracting from the additive part through coring and cutting. The results of the experiment show that underwater 3D printing required a greater amount of printing output than in-air 3D printing for a good print quality, and buildability was improved underwater compared to that in air. In the case of the specimen layered underwater, the density and compressive strength decreased compared to the specimen layered in air. Because there are almost no effects of moisture evaporation and bleeding in water, the interlayer bond strength of the specimen printed underwater was somewhat larger than that printed in air, while there was no effect of the deposition time interval underwater.

## 1. Introduction

The development of 3D printing technology has made such a great contribution to the manufacturing industry that it is often called the Fourth Industrial Revolution. The Contour Crafting research team began developing 3D printing equipment suitable for various materials in the late 1990s and devised a 3D concrete printing (3DCP) technique for construction in the early 21st century [1,2]. The developed 3DCP for construction mainly applies the additive manufacturing (AM) method by material extrusion [3,4]. Another 3DCP method is a powder bed-based method that performs selective binding through binder jetting [5,6].

The 3DCP technique has many advantages over traditional construction methods for concrete structures. It eliminates the formwork process and subsequent compaction process, which can shorten the construction time and minimize waste and labor. The 3DCP technique can increase the precision of the structure, and because it allows high degrees of freedom in shape, free-formwork construction of irregular structures can be performed on site without any restrictions on location [7,8,9,10].

Various concrete structures have been constructed using 3DCP technology [11]. A two-story municipality building in Dubai by Apis Cor [12], a community village in Austin by ICON [13], vehicle-hiding concrete arches in California by the U.S. Marines Corps [14], a prestressed bicycle bridge in Eindhoven [15], and optimized reinforced concrete beams in Ghent and Naples [16,17] were constructed by the 3DCP technology.

Despite these advantages and applications, 3DCP still faces many challenges. One of them is to design a printing material, concrete, which is suitable for the structure to be applied and compatible with the equipment to be used. Printing materials must meet various requirements for pumpability, printability, buildability, and open time in an unhardened fresh state [18]. These requirements, however, have characteristics that both conflict with each other and vary depending on variables such as place, equipment, and environment, thus designing an optimized printing material in a short period of time is difficult and is still a major research subject for 3DCP that remains unsolved. Printing materials must satisfy performance requirements in terms of mechanical properties such as compressive strength, interlayer bond strength, and durability in the hardened state. However, the performance of hardened concrete varies depending on the method used to make the test specimen, test method and condition, specimen size, etc. [19]. Furthermore, there is still no standard for evaluating and judging the material requirements and characteristics.

Recently, the idea of applying 3DCP technology to offshore construction was proposed. The idea of using underwater additive manufacturing, where the effect of gravity is decreased and freedom of form is greater, seems attractive [20]. Underwater 3DCP can be applied to the maintenance of damaged underwater structures. Underwater structures with complex shapes depending on the purpose of use, such as wave-dissipating blocks and reef blocks, have limitations in manufacturing them with formwork; however, the underwater 3DCP technology will enable optimized underwater structure production [21].

This study examined the feasibility of these new ideas. The same mixture of mortar was 3D printed in air and underwater, and the difference was examined. In the fresh state, the print quality and shape stability were compared. In the hardened state, the mechanical properties of density, compressive strength, interlayer bond strength, flexural tensile strength, and splitting tensile strength were evaluated and compared.

## 2. Materials and Methods

### 2.1. Materials

Taking the 3DCP machine conditions into consideration, a mortar with no coarse aggregates was used as the printing material. The specified compressive strength was set to 50 MPa, and a mixture with a water-to-binder ratio of 38.4% was used, as shown in Table 1. The mixture proportioning for 3D printing was obtained through our own multiple mixing and printing tests.

A type 1 ordinary Portland cement with a specific gravity of 3.15 g/cm^3^, specific surface area of 3770 cm^3^/g, and setting times of 210 min for the initial setting and 290 min for the final setting was used. To achieve high strength and durability characteristics through the filling of voids between cement particles, 10% of the binder amount was substituted with an undensified silica fume. As a result of testing the used silica fume according to ASTM C1240 [22], the SiO_2_ content was 91.6%, loss on ignition was 2.3%, 45 μm residue was 3.9%, and specific surface area was 204,000 cm^2^/g. Silica sand with a SiO_2_ content of 95.5% was used as a fine aggregate to maintain a constant particle size control. To satisfy the shape stability of the printed materials in 3D printing, the particle size distribution of fine aggregates was artificially adjusted; thus, the ratio of No. 3, 6, and 7 silica sand with sizes of 1.2~2.4 mm, 0.25~0.70 mm, and 0.17~0.25 mm, respectively, was set to 3:4:3.

To improve the buildability of the mortar after 3D printing and to provide anti-washout performance underwater, a viscosity-modifying agent (VMA) produced by Dongnam was used in 2.0% of the water content. A powder-type cellulosic VMA with a density of 0.75 ± 0.05 g/cm^3^ and a solid content of 97.0 ± 2.0% was used. In order to extrude and print the mixed mortar smoothly through the 3DCP machine without clogging, 0.8% of a high-range water-reducing agent (HRWRA) compared to the binder contents was applied. A light brown liquid polycarboxylic acid-based HRWRA, produced by Dongnam, with a density of 1.07 ± 0.1 g/cm^3^ and a solid content of 30.1 ± 2.0% was used.

Mortar mixing based on the mixture proportioning of Table 1 was performed twice, one for 3D printing in air and the other for 3D printing underwater. Two mixings and two 3D printing tests were carried out separately in the same order. This is because the mechanical properties of 3DCP can be influenced by the age of the fresh mortar mixture [19]. After mixing each mortar, the slump flow test of ASTM C1437 [23] was performed immediately after mixing and at the start of 3D printing to measure the consistency of the mortar mixture. While the hydration of the cementitious particles is not taken into consideration, buildability and pumpability are closely related to the slump flow of the material [24]. As a result of the slump flow test, the slump flow of the mortar immediately after mixing and at the start of 3D printing was 112.3 mm and 124.3 mm, respectively, in air, and 112.2 mm and 121.1 mm, respectively, underwater. These are smaller values than the values of 150~190 mm suggested by Tay et al. [24] as the optimal condition for good buildability and a smooth surface. However, this slump flow can vary depending on various conditions such as the printing material, mixing proportioning, and 3DCP machine. The slump flow at the start of 3D printing increased by approximately 10 mm compared to the time immediately after mixing. This was probably because heat was generated as the mortar passed through the rotor stator and hopper, causing the HRWRA to react more actively. The consistency of the fresh mortar between the 3D printing test underwater and in air was similar, confirming that the basic material printing conditions were maintained the same.

### 2.2. 3D Concrete Printing Machine

For the application of 3DCP, the gantry robot type of underwater additive layering machine manufactured by our team as shown in Figure 1 was used. The gantry-type equipment operates on the X, Y, and Z axes with respective stroke ranges of 2500, 1200, and 1500 mm through a linear guide rail. Additionally, it controls the movement speed, movement distance, and rotation angle of the axis using a servo motor and a speed reducer. A worm pump method using a rotor/stator was used for the concrete pumping and conveying equipment. With this method, the fresh mortar is filled into the reservoir of the pump and, when pumped by the rotor/stator, moves to the hopper of the 3D printing device through a circular hose. The spindle shaft installed on the top of the hopper in the 3D printing equipment transfers the printing material to the tip of the nozzle at a constant amount and constant speed while the concrete is agitated secondarily through a screw auger. The transferred concrete is then printed through a nozzle. The nozzle has a square opening of 59.5 (*n_w_*) × 28.5 mm (*n_t_*), with fixed finishing blades of 28.5 (*n_t_*) × 28.5 mm (*h_b_*) installed on both sides of the tip of the square nozzle, as shown in Figure 1. The 3DCP machine is controlled using an MXP 2.0 N-Type motion controller. This module basically supports EtherCAT communication and various topologies, supports servo control for up to 32 axes and 9 simultaneous command axes, and supports the PLC program and G-code program, a computer numerical control function. The water tank was made of transparent polycarbonate, and the dimensions in the X, Y, and Z axes were 1600 × 1200 × 550 mm.

### 2.3. Test Specimen

#### 2.3.1. Specimens Produced by Direct Casting in Cylindrical Molds

To examine the characteristics of the mortar that was printed before manufacturing the additive parts, cylinder specimens were fabricated in air and underwater. The nozzle of the 3DCP machine was placed as far down as possible inside a cylindrical mold with a diameter of 100 mm and a height of 200 mm; thereafter, the 3D printing machine was operated, and the mortar was printed out. At the same time, the nozzle was gradually moved upward to ensure the mortar would sufficiently fill the mold. The printed mortar sample was added until the sample was in the shape of a hill on the top of the mold (Figure 2). After compacting the sample filled in the mold using a tamping rod and rubber mallet, the sample above the mold was removed, and the surface was carefully finished. After that, to prevent drying of the surface, the specimens were covered with plastic, demolded after two days, and moved to a curing water tank. In the case of manufacturing specimens by direct casting underwater, the same process through the input of the printed mortar sample was carried out underwater, and then compaction was carried out in air (Figure 2). The mold filled with the sample was gently taken out from the water and compacted; the surface was finished, and then moved to a water tank for curing. At 2 days of age, the mold was taken out from the curing tank, demolded, and immediately re-cured in water.

To apply 3DCP, a very stiff mixture with a very small slump is used to ensure the buildability of the printed mixture. If compaction is not performed properly in this stiff mixture, the concrete will not be filled well and will have a large void inside, adversely affecting the strength and durability of concrete structures [25]. In this study, compaction was performed using a tamping rod and a rubber mallet according to the compaction method of ASTM C31 [26], but problems such as difficulty in compaction after one-step full casting and the addition of water to the mixture by compaction after being cast underwater emerged. Therefore, to examine the differences in the characteristics of cylindrical specimens due to the presence or absence of compaction by tamping rods, specimens (M-O) with both tamping rod compaction and rubber mallet compaction and specimens with only rubber mallet compaction (M-X) were prepared. As shown in Table 2, the specimens manufactured by direct casting in cylindrical molds were used for compressive strength and splitting tensile strength tests.

#### 2.3.2. Additive Manufacturing of Parts

The additive manufacturing of 3DCP parts was carried out both in air and underwater. The laboratory temperature and humidity were 25 °C and 61%, respectively, and the temperature of the water in the water tank was 23 °C. As shown in Figure 3, Figure 4 and Figure 5, all parts were printed in a 1 m-long linear shape, and all layers were printed in the same direction to maintain the same time gap between layers. The printing height of each layer was set to 30 mm. 

In the 3D printing test in air, two parts of four layers and two layers, AP-4La and AP-2La, respectively, were fabricated in order (Table 2, Figure 3). The 4-layer part (AP-4La) was used for coring the compressive strength specimens, and the 2-layer part (AP-2La) was used to cut the specimens for flexural tensile strength, compressive strength, and interlayer bond strength testing. The rotation speed of the spindle shaft in the hopper was set to 15 rpm, which corresponds to a printing volume of approximately 87 Ml/s. The nozzle movement speed was 2500 mm/min, and the printing time gap between the layers was approximately 50 s.

In the 3D printing test underwater, one 4-layer and two 2-layer parts were fabricated at a water depth of 2200 mm (Table 2, Figure 4). The 4-layer underwater part, WP-4La, was used for coring the compressive strength specimen, and the 2-layer underwater parts, WP-2La and WP-2La-15, were used to cut the specimens for flexural tensile strength, compressive strength, and interlayer bond strength testing. For the WP-4La and WP-2La parts, the conditions of a nozzle movement speed of 2500 mm/min and an interlayer time gap of 50 s in the in-air 3D printing test were equally applied, but for the WP-2La-15 part, an interlayer time gap of 15 min was applied. This was conducted to examine the difference in mechanical properties according to the difference in the printing time interval between layers underwater. In the first attempt of the underwater additive experiment, the experiment was performed under the same printing speed conditions as in the air additive experiment. However, as shown in Figure 5, surface defects and discontinuities appeared in the printed part, resulting in a poor print quality. It seems that there are many causes for these such as an increase in the viscosity between the mortar and the finishing blade of the nozzle in the water. Further research is required to determine the cause of this problem. To provide a stable printing quality, the rotation speed of the spindle shaft in the hopper was set to 19 rpm, and the parts shown in Figure 4 were manufactured underwater.

#### 2.3.3. Specimens Produced by Extracting from Parts

Compressive strength specimens were produced by coring the AP-4La and WP-4La parts with 4 layers underwater and in air before the parts hardened. For the coring mold, a cylindrical mold with a diameter of 50 mm and a height of 100 mm made of coated paper was used. The closed bottom face was cut off, opening the top and bottom faces of the mold, making the coring operation easier. The process of producing compressive strength specimens by coring additive parts is shown in Figure 6. After about 20 min of fabricating the parts, the coring mold was penetrated vertically. Thereafter, the specimens were separated by removing the remaining parts outside the mold before they hardened. Specimens produced in air were cured in air for two days with plastic applied and, after demolding, were placed in curing water. In the case of specimens produced underwater, after separating the specimens from the part, the specimens were transferred to a curing water tank, taken out after two days, demolded, and cured in water again.

Specimens for flexural tensile strength, compressive strength, and interlayer bond strength tests were prepared by cutting the AP-2La, WP-2La, and WP-2La-15 parts printed in two layers in air and underwater. Before the parts hardened, approximately 30 min after the parts were manufactured, the parts were cut in advance for each strength test purpose, as shown in Figure 7, and curing was started. In the case of parts manufactured in air, water curing was started two days after the plastic application in air. The parts manufactured underwater were cured in the water tank where they were manufactured for two days and then transferred to a curing water tank to continue the water curing.

After curing, the test specimens were manufactured by processing the cut parts. The specimens for the flexural tensile strength test were processed by cutting both sides to a width of 60 mm using a diamond saw. After the flexural tensile strength test, the two portions of the broken prisms were used as specimens for the compressive strength test. The test specimens for the interlayer bond strength test were cut with a saw on six sides such that the width × length × height was 60 × 50 × 55 mm. The interlayer was notched such that the cross-section was approximately 30 × 25 mm. 

### 2.4. Test Method

The density of the 50 by 100 mm and 100 by 200 mm cylindrical specimens at 7 days of age and 28 days of age was measured according to EN 12390-7 [27]. The mass used for the density calculation was measured with a scale in a surface dry condition in air, and the volume was obtained by actual measurements. Three specimens per variable were measured for density evaluation.

The compressive strength of 50 by 100 mm and 100 by 200 mm cylindrical specimens at 7 and 28 days of age was measured according to ASTM C39 [28] (Figure 8a). The compressive strength test was performed using a 5 MN compression tester from SHIMADZU at the Intelligent Construction System Core-Support Center of KBSI, and a load was applied at a rate of 0.25 MPa/s. At the age of 7 days, two displacement transducers were installed, and the static modulus of elasticity was measured according to ASTM C469 [29]. Three cylindrical specimens per variable were tested for compressive strength evaluation.

Figure 9 shows the details of the flexural strength and compressive strength tests of the specimens extracted from the parts. In both tests, a load was applied using a 5 MN UTM from SHIMADZU in the direction perpendicular to the direction of printing at the age of 7 days (Figure 8b). In many studies [4,30,31,32], performance evaluation was performed in three directions considering the anisotropy of 3DCP, but in this study, only one direction was tested due to the limitations of specimen production. The flexural strength test on the specimens extracted from the parts was performed in accordance with ASTM C348 [33] using a three-point loading method (Figure 8a). However, the width and span length of the flexural test specimen were 60 and 150 mm, respectively, which were different from the 40 and 100 mm, respectively, suggested in ASTM C348 [33]. The ratio of the width to the span length of 2.5 was kept the same. The load was applied at a rate of 0.12 MPa/s. The compressive strength test on specimens extracted from parts was conducted on the broken prism halves resulting from specimens tested in the flexural tests according to ASTM C349 [34] and EN 1015-11 [35] (Figure 8c and Figure 9b,c). However, the width of the specimen and the width of the load plate were 60 mm, which deviated from the 40 mm suggested in ASTM C349 [34] and EN 1015-11 [35]. The load was applied at a rate of 0.25 MPa/s. Three flexural tensile strength and six compressive strength tests per variable were performed.

Because the interlayer bond strength is one of the important factors determining the safety of 3DCP structures, many studies have been conducted on this property. Most researchers [32,36,37,38] evaluated the interlayer bond strength through a pull-out test in which direct tension was applied, and as shown in Figure 10, this study performed an experiment similar to the pull-out test performed by previous researchers. Two metallic brackets were epoxy glued to the top and bottom of the notched specimen. Delatte et al. [39] stated that the pull-out experiment largely depends on the eccentricity of the applied load. To minimize the effect of such eccentricity, ring-shaped shackles that can rotate in both directions were mounted on the upper and lower parts of the specimen. The interlayer bond strength test was conducted under displacement control at the rate of 1 mm/min at 28 days of age using a 500 kN UTM from MTS. Four specimens per variable were tested for interlayer bond strength evaluation.

Splitting tensile strength tests on 100 by 200 mm cylindrical specimens were performed according to ASTM C496 [40] using a 5 MN UTM from SHIMADZU. Three 28-day-old specimens were used for each variable, and the tests were performed at a load rate of 1.0 MPa/min.

## 3. Results and Discussion

### 3.1. Printability and Buildability

The nozzle movement speed and the spindle shaft rotation speed, which determine the amount of mortar output, applied during printing additive parts, were determined through print quality evaluation. For the printing quality to be considered acceptable, there should be no surface defects or discontinuity of the printed layer, the layer edges should be visible and squared, and dimensional consistency should be satisfied [41]. In order to achieve a good print quality that can satisfy these three conditions, repeated print tests were conducted by changing the printing conditions in air in advance, thus determining the nozzle movement speed of 2500 mm/min and spindle shaft rotation speed of 15 rpm. As mentioned in Section 2.3.2, when the same printing conditions as those when fabricating parts in air were applied to the production of additive parts underwater, surface defects and discontinuity of the printed layers appeared, and the printing condition was rejected. By changing the spindle shaft rotation speed from 15 to 19 rpm, a good print quality was achieved.

To evaluate the buildability of the printed layers, the width and height of each layer of the additive parts were measured (Figure 11). Because measuring the dimensions of the parts in the fresh state underwater was difficult, each layer’s dimensions were measured using a caliper after the parts hardened. As shown in Figure 11, the width of each layer increased, and the height decreased as it went to the lower layer of each part. In particular, this trend increased as the number of layers in the part increased. This is due to the weight of the following layers and the extrusion pressure when printing them [41,42]. To achieve good buildability, the lower layer should resist the weight and extrusion pressure of the subsequent layers and maintain its original shape well. The layer height of the additive parts printed underwater was higher than that printed in air. This is because the vertical load acting on the lower layer by the weight of the upper layer and by the printing pressure is reduced due to buoyancy underwater. Normally, if the layer height increases and the same amount of print output is used as that in air, the layer width should decrease, but instead, the layer width also increased underwater, due to the increase in the print output amount needed to satisfy the requirement of a good print quality during additive manufacturing underwater. If the print output amount is the same, it is thought that the change in the height and width of the layer is reduced and the buildability is improved underwater rather than in air. In the case of the WP-2La-15 part, in which the interlayer time gap was increased to 15 min, the change in the height and width of the lower layer was smaller than that of the WP-2La part. This is because the yield strength of the lower layer increased for 15 min and resisted the weight and pressure of the upper layer. However, the buildability was not yet satisfied as the dimension of the lower layer decreased by 10–15%. Therefore, the age of the printed mortar and the time gap between layers are very important for a good buildability of 3DCP. Given this, the material and equipment conditions should be optimized when 3DCP is applied to a structure. For a deeper understanding of the printability of the printing mixture in terms of buildability, further fundamental research on the rheology and early age mechanical behavior of printed concrete, especially underwater, is necessary [41,43,44,45]. 

### 3.2. Compaction Method Difference of Specimens Produced by Direct Casting in Cylindrical Molds

Specimens were manufactured by direct casting in cylinder molds to understand the characteristics of the printed mortar itself before the production of the additive parts. Differences in properties based on the presence or absence of tamping rod compaction when manufacturing cylindrical molds were investigated using 7-day-old specimens. The differences in properties in terms of surface quality, density, compressive strength, and elastic modulus were examined, and the results are shown in Table 3.

Figure 12 shows the specimens produced by direct casting in cylindrical molds after demolding. As shown in the figure, all specimens had some surface faults. This was because the material used was a very stiff mixture. Quantitative comparison of surface faults was not performed, but visual observation determined that the specimen without tamping rod compaction (-X) had more surface faults than the specimen with tamping rod compaction (-O), and the specimen cast in air (AP) had more surface faults than the specimen cast underwater (WP). The tamping rod compaction process appears to fill the surface voids better because the compaction and water reduce the friction between the mold and the material.

The apparent density was measured to compare not only surface faults but also internal pores. As shown in Table 3, the specimen without tamping rod compaction (-X) had a slightly larger density by approximately 1.3% than that with tamping rod compaction (-O), and the specimens cast in air (AP) and the specimens cast underwater (WP) had the same density. This result contradicts the visual observation of the surface and leads to the deduction that the degree of internal pores is different from that of the surface faults. As a result, it can be concluded that there is almost no difference in the pores of the specimen due to the presence or absence of tamping rod compaction or casting underwater or in air.

In the case of the compressive strength of 7-day-old specimens, the specimens without tamping rod compaction (-X) were 6.5% higher in air and 1.0% higher underwater than the specimens without tamping rod compaction (-O). The specimens cast underwater (WP) showed no difference or a 5.2% lower compressive strength than the specimens cast in air (AP). In the case of the modulus of elasticity of 7-day-old specimens, the non-tamping rod compaction specimens (-X) were 2.7% higher in air and 3.1% lower underwater than the tamping rod compaction specimens (-O). The specimens cast underwater (WP) showed a modulus of elasticity that was only 4.4% higher with tamping rod compaction and 1.5% lower with no tamping rod compaction than the specimens cast in air (AP). By substituting the measured density and compressive strength into the formula for calculating the elastic modulus in the ACI 318 code [46], the elastic modulus was calculated and was found to be approximately 20% higher than the measured elastic modulus. This might be because the material used is a mortar that does not contain coarse aggregates, meaning deformation occurs more easily. In conclusion, the compressive strength and modulus of elasticity did not show trends according to the presence or absence of tamping rod compaction, and to the in air or underwater variables; even if there were trends, they were found to be at the standard deviation level. Figure 13, which shows the stress–strain relationship curves for all specimens produced by direct casting in cylindrical molds, also shows no significant difference between the variables. 

### 3.3. Density

The average density of specimens produced by direct casting in cylindrical molds was 2091 kg/m^3^ at the age of 7 days and 2103 kg/m^3^ at the age of 28 days, and it was found that there was no difference between the specimens cast underwater and in air within a 1% difference (Figure 14). The density is higher than that of ordinary mortars (1800 kg/m^3^) and sprayed mortars (1800–2000 kg/m^3^) [47], but smaller than that of Le et al.’s mixture [48]. This is mainly due to the water-to-binder ratio and cement content. Specimens prepared by coring additive parts showed a difference in density between the additives in air and underwater. The density of the specimens produced by coring additive parts in air (AP-CO) was 2096 kg/m^3^ at the age of 7 days and 2128 kg/m^3^ at the age of 28 days, similar to the density of specimens produced by direct casting in cylindrical molds. However, the density of specimens produced by coring additive parts underwater (WP-CO) showed a decrease of approximately 2.5% compared to that in air, indicating that grading and homogeneity were somewhat lower than those of the in-air additive part. In all specimens, an increase in density was observed with increasing age. It seems that the additional production of hydrates by the hydration contributed to the increase in density. The density of the 28-day-old specimens compared to the 7-day-old specimens increased by 0.2–0.9% for specimens produced by direct casting in cylindrical molds and by 1.6–1.7% for specimens produced by coring additive parts.

### 3.4. Compressive Strength

Figure 15 shows the results of the 7-day and 28-day compressive strength tests. Overall, the compressive strength of the specimens extracted from the parts was lower than that of the specimens produced by direct casting in cylindrical molds. This is because the compressive strength is reduced as the original mortar is printed and additively layered without restraint and compaction of the form. Considering that the size or aspect ratio of specimens extracted from parts is smaller than that of specimens produced by direct casting, the reduction in compressive strength may be somewhat greater. In general, the smaller the size of the specimen and the smaller the aspect ratio, the larger the measured compressive strength of the concrete [49]. Compared to the compressive strength of the original mortar before 3D printing, the compressive strength after additive manufacturing decreased by 23–43% for the 7-day-old specimen and 13–21% for the 28-day-old specimen. Research by Lee et al. [50] showed a greater reduction in compressive strength; the additively manufactured specimens’ compressive strength was approximately 30% that of the 28-day-old mold-casted specimens. This difference in compressive strength may vary depending on factors such as printing pressure, print quality, and buildability. For the increase in the rate of compressive strength according to age, that of the specimens produced by coring from parts was more than twice that of specimens produced by direct casting. This trend was similar to the density results. This is thought to be because the hydration rate slowed under the reduced external restraint. Specimens produced by extracting from parts showed a greater deviation in compressive strength than specimens produced by direct casting. This means that the internal structure of the matrix is not kept constant along the parts because it is layered without confinement and compaction in the formwork, and it is thought that the deviation gradually decreases as age increases.

Specimens obtained by direct casting in cylindrical molds did not show any difference in compressive strength between underwater and in-air casting. On the other hand, the specimens extracted from the parts showed a difference in compressive strength, and the compressive strength of the specimen printed underwater was lower than that printed in air. The reason is thought to be that the decrease in the self-weight and extrusion pressure caused by water slowed the hydration rate. The difference in the compressive strength of the specimens printed underwater and in air may hardly appear after the age of 28 days. The compressive strength of the WP-CO specimen produced by coring the additive part was lower than that of the AP-CO specimen by 21% at the age of 7 days and by 6% at 28 days. The WP-CU specimen produced by cutting the additive part showed a 16% lower compressive strength than the AP-CU specimen at 7 days. The compressive strength of the specimen produced by cutting the parts measured only at the age of 7 days showed a similar trend to that of the specimen produced by coring the parts. Thus, the method of coring the parts can be considered one method of measuring the compressive strength of an actual 3DCP structure.

### 3.5. Flexural Tensile Strength

The flexural tensile test results are presented in Figure 16. Among the dry-sprayed mortars in the research of Austin et al. [47], the flexural tensile strength of the specimen with a similar compressive strength (38.8 MPa) and density (1973 kg/m^3^) to the specimens in this study was 6.2 MPa. 

In the case of the AP-CU specimen printed in air, a similar flexural tensile strength value was observed. The flexural tensile strength of WP-CU and WP-CU-15 specimens printed underwater increased by approximately 30% compared to that of the AP-CU specimens. This result is the opposite of the density and compressive strength test results. The reason can be ascribed to the anisotropic properties of 3DCP. In the longitudinal direction where the principal tensile stress acts, the confining force acts the least compared to other perpendicular and lateral directions, and the longitudinal confining force is mainly determined by the amount of print output and the speed of the nozzle movement [4]. The WP-CU and WP-CU-15 specimens layered underwater have a larger amount of print output than the AP-CU specimen layered in air, meaning the longitudinal confining force is larger; thereby, it can be presumed that the flexural tensile strength of the specimen layered underwater was larger than that of the specimen layered in air. However, it is necessary to find out the exact cause of this difference in the flexural tensile strength through detailed additional research.

The WP-CU-15 specimen showed a flexural tensile strength similar to that of the WP-CU specimen. This is because the lower layer, where the maximum tensile stress is applied, was additively layered under the same condition as the two specimens, and it was independent of the difference in the time interval between layers.

### 3.6. Interlayer Bond Strength

As shown in Figure 17, the 28-day interlayer bond strength of all specimens showed similar results of 2.43–2.57 MPa regardless of the variable. These values are very high, comparable to the values of around 2.4 MPa calculated through the concrete direct tensile strength equation proposed by Kim and Taha [51]. In addition, the interlayer bond strength results were within the range of those shown in the study by Le et al. [18]: 1.8–2.8 MPa at printing time gaps of 0 and 15 min. The interlayer bond strength of the specimens additively layered in air (AP-CU) was about 5–6% lower than that of the specimens layered underwater. It is thought that evaporation of moisture from the surface of the first printed layer at 50 s short time intervals between layers influenced this result [32]. Unlike additive printing in air, underwater additive printing seems to have an advantage in terms of reducing the interlayer bond strength due to water evaporation.

There was no difference in the interlayer bond strength according to the interlayer deposition time interval in water. In the study of Marchment et al. [32], the interlayer bond strength decreased due to surface moisture evaporation until the additive deposition time interval of about 20 min, and thereafter, the interlayer bond strength increased again due to the bleeding of the mortar in the lower layer. However, it was found that there was no change in the interlayer bond strength according to the deposition time interval in this study because the effects of water evaporation and bleeding were excluded by the water in the water tank at the 15 min underwater deposition time interval. According to the study of Keita et al. [52], when the first layer is protected from drying, the interlayer bond strength remains constant with increasing resting time between layers up to 2 h. It is necessary to validate this result through additional studies that subdivide the deposition time interval underwater.

### 3.7. Splitting Tensile Strength

The splitting tensile strength was measured using specimens cast directly into molds at the age of 28 days. As shown in Figure 18, the specimens cast underwater showed approximately 5% higher average values and larger deviations than the specimens cast in air. These results are similar to the 28-day compressive strength results of the specimens cast directly into the mold. Through this, it was evaluated that there was no significant difference in the mechanical properties of the mortar cast in air vs. underwater in the state constrained by the mold. The reason why the compressive strength and splitting tensile strength of the specimens cast underwater are rather high is thought to be due to the difference in the curing conditions of the first two days [49].

## 4. Conclusions

In this study, the same mortar mixture was printed and additively layered in air and underwater, and the properties in the fresh and hardened states were compared. The specimens were produced by direct casting into the mold and by extracting through coring and cutting after manufacturing the additive part. In the fresh state, the printability and buildability were evaluated, and in the hardened state, the mechanical properties of density, compressive strength, interlayer bond strength, and flexural and splitting tensile strengths were evaluated. Based on the test results and discussion, the following conclusions can be drawn:

(1) As a result of 3D printing underwater with the same printing conditions as printing in air, many defects and discontinuities occurred; therefore, a greater amount of printing output was required in 3D printing underwater than in 3D printing in air.

(2) The reduction in the layer height of the part decreased due to the reduction in the weight and pressure underwater compared to that in air. By increasing the time gap between layers to 15 min, the decrease in the layer height further decreased and buildability was improved.

(3) In the specimens produced by direct casting in cylindrical molds, there was no difference based on the presence or absence of tamping rod compaction in terms of density, compressive strength, and elastic modulus.

(4) The density of the part printed underwater was lower than that of the part printed in air, and the density increased with increasing age.

(5) Specimens extracted from parts showed a lower compressive strength than specimens produced by direct casting into cylindrical molds because the material was additively layered without confinement of the formwork. In addition, due to the decrease in weight and pressure underwater, the compressive strength development of the part was slower underwater than in air.

(6) Because there is almost no effect of moisture evaporation and bleeding in water, the interlayer bond strength of the specimen printed underwater was somewhat larger than that printed in air, and there was no effect due to the deposition time interval underwater.

(7) Additive layering underwater was evaluated to be more advantageous than that in air in terms of the flexural tensile strength.

These experimental results were analyzed within the possible range based on existing theories and previous research. Some points that have not yet been clearly identified will be analyzed in depth through additional research.

## Figures and Tables

**Figure 1 materials-14-05888-f001:**
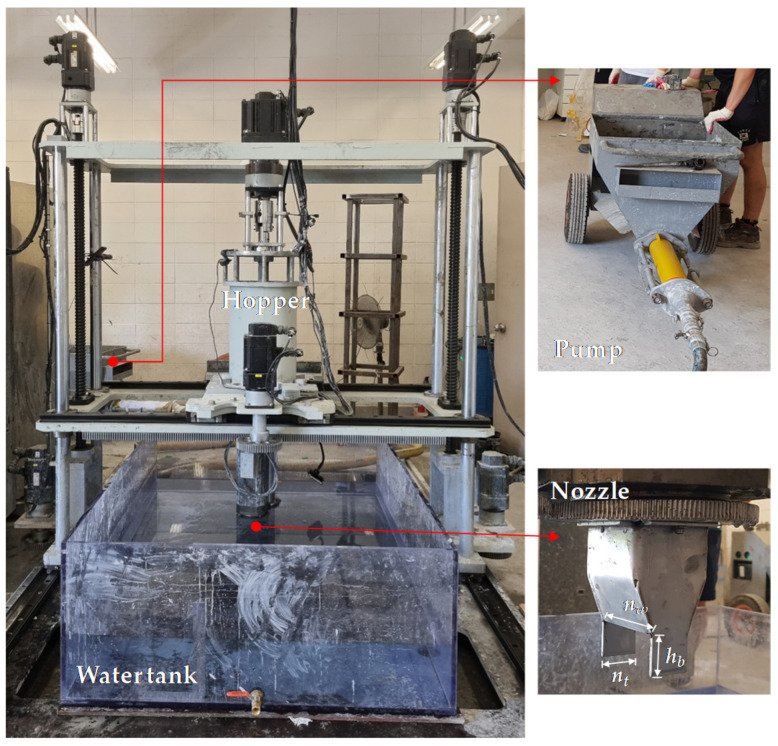
3D concrete printing machine.

**Figure 2 materials-14-05888-f002:**
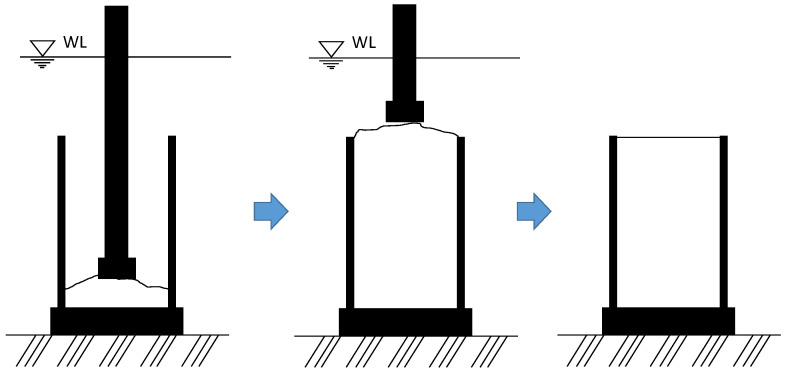
Manufacturing method of cylindrical specimen by direct casting of 3D-printed mortar.

**Figure 3 materials-14-05888-f003:**
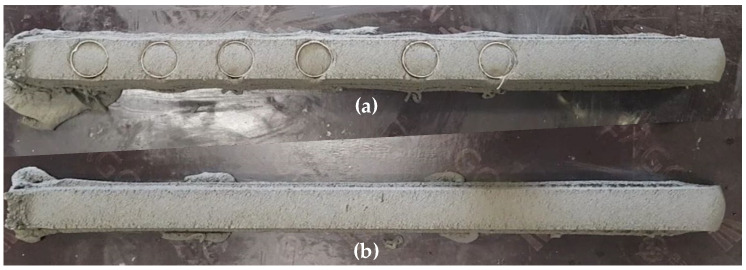
Parts additively manufactured in air: (**a**) AP-4La; (**b**) AP-2La.

**Figure 4 materials-14-05888-f004:**
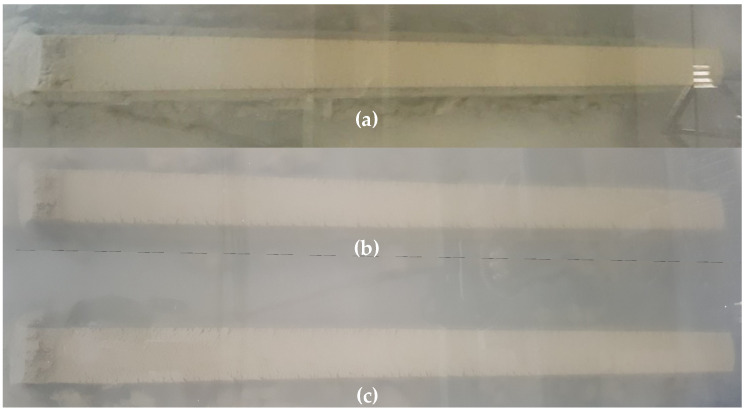
Parts additively manufactured underwater: (**a**) WP-4La; (**b**) WP-2La; (**c**) WP-2La-15.

**Figure 5 materials-14-05888-f005:**
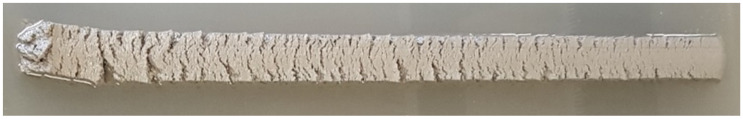
Parts additively manufactured underwater with a spindle shaft rotation speed of 15 rpm.

**Figure 6 materials-14-05888-f006:**
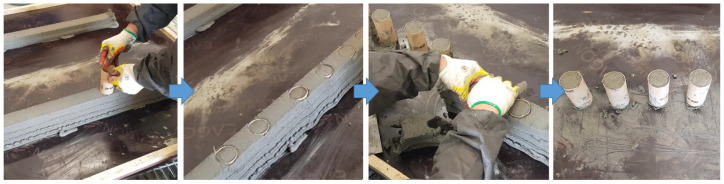
Production process of specimens by coring additive parts.

**Figure 7 materials-14-05888-f007:**
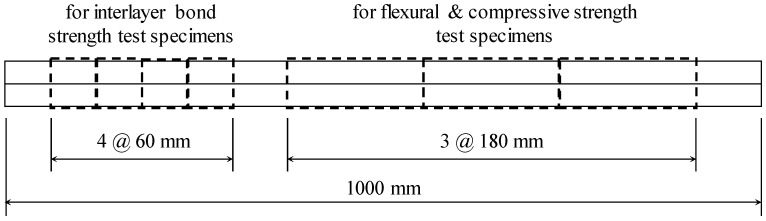
Cutting of additive parts before hardening.

**Figure 8 materials-14-05888-f008:**
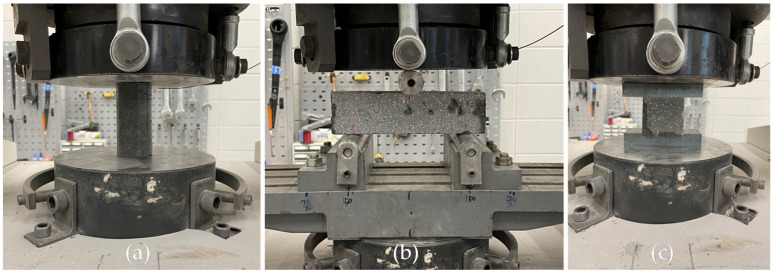
Scenes of mechanical test: (**a**) compressive strength test for 50 by 100 mm cylindrical specimens; (**b**) flexural tensile strength test for specimens cut from parts; (**c**) compressive strength test for specimens cut from parts.

**Figure 9 materials-14-05888-f009:**
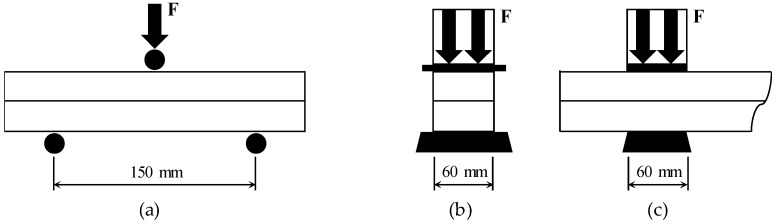
Details of flexural and compressive strength tests using specimens extracted from parts: (**a**) front view of flexural test; (**b**) side view of compressive strength test; (**c**) front view of compressive strength test.

**Figure 10 materials-14-05888-f010:**
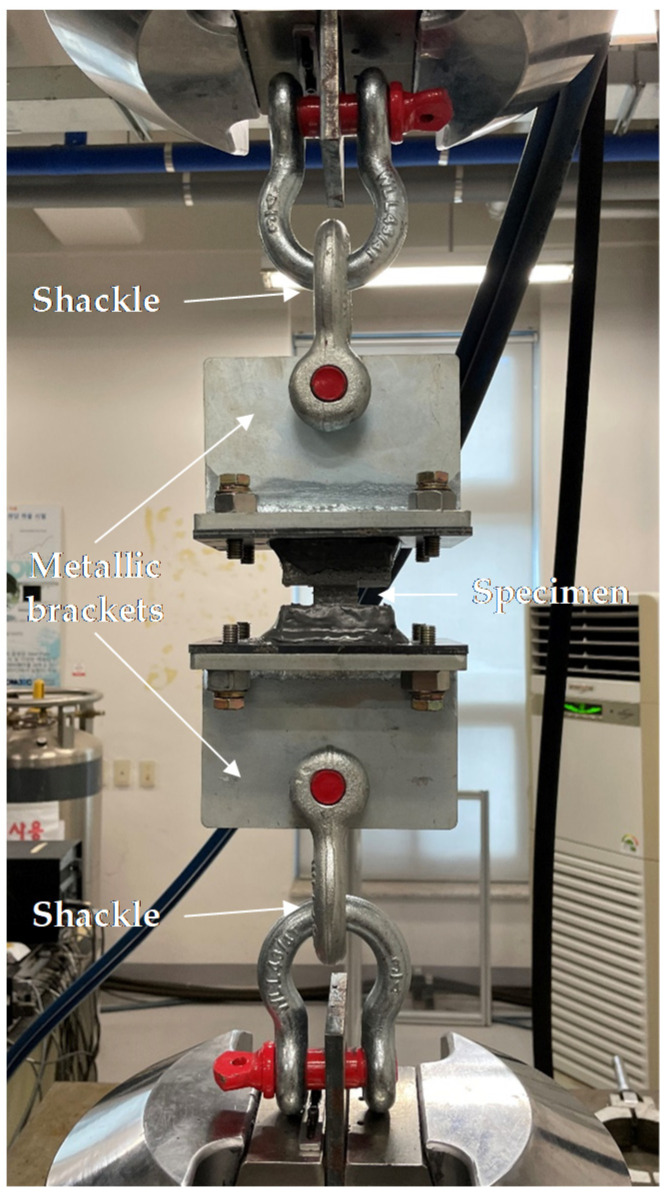
Interlayer bond strength test.

**Figure 11 materials-14-05888-f011:**
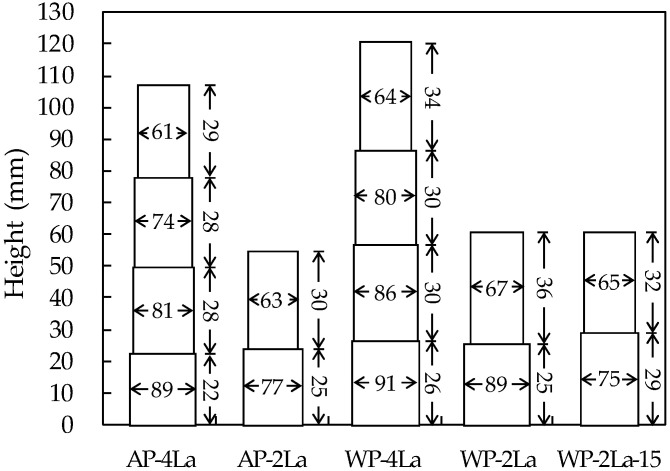
Measurement of height and width for each printed layer of parts.

**Figure 12 materials-14-05888-f012:**
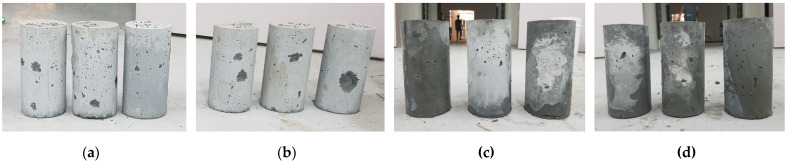
Specimens produced by direct casting in cylindrical molds after demolding: (**a**) AP-M-O; (**b**) AP-M-X; (**c**) WP-M-O; (**d**) WP-M-X.

**Figure 13 materials-14-05888-f013:**
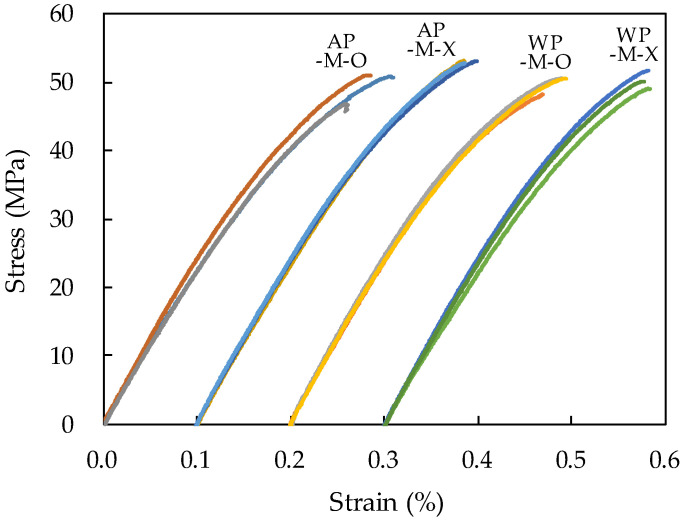
Stress–strain relationship of specimens produced by direct casting in cylindrical molds (the strain in all graphs starts from 0).

**Figure 14 materials-14-05888-f014:**
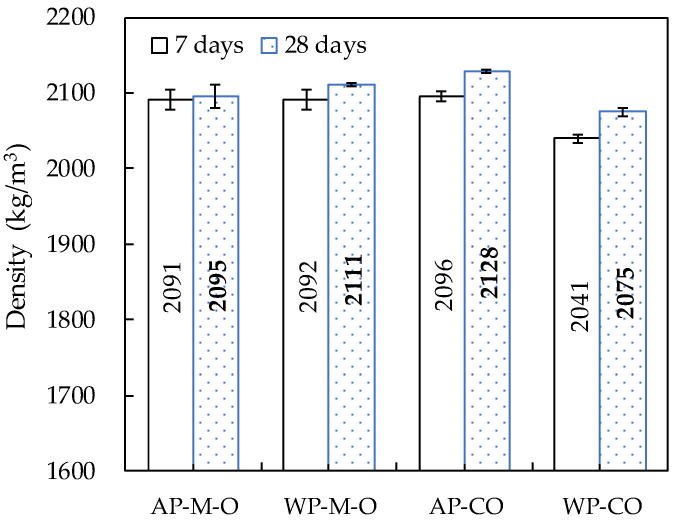
Density of cylindrical specimens.

**Figure 15 materials-14-05888-f015:**
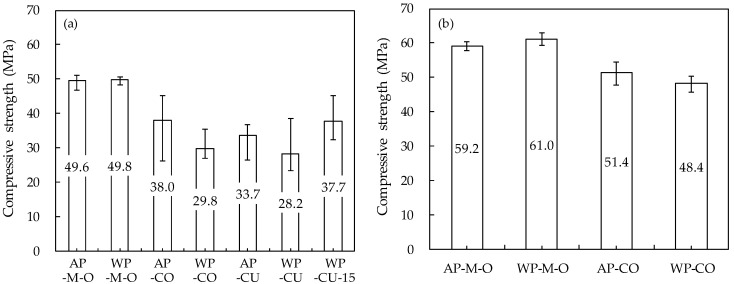
Compressive strength: (**a**) 7 days of age; (**b**) 28 days of age.

**Figure 16 materials-14-05888-f016:**
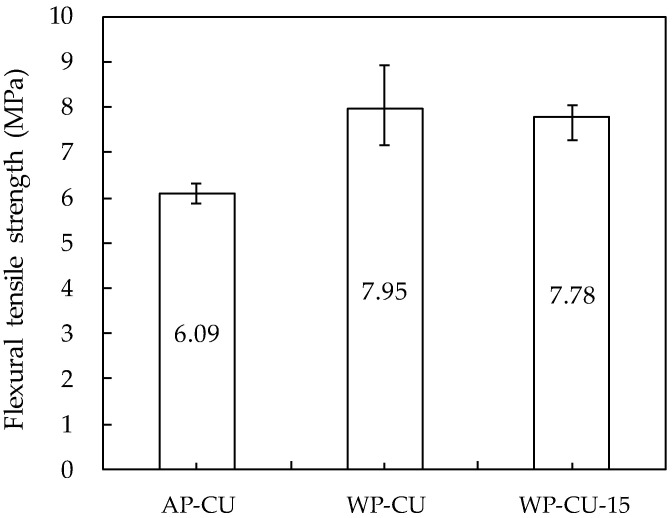
Flexural tensile strength.

**Figure 17 materials-14-05888-f017:**
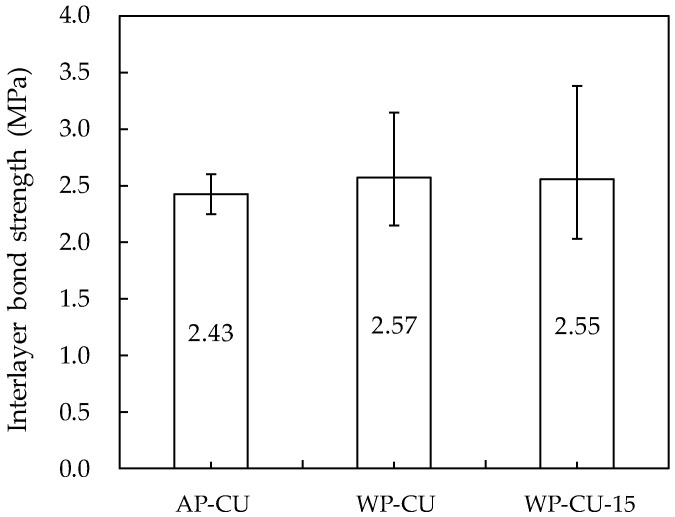
Interlayer bond strength.

**Figure 18 materials-14-05888-f018:**
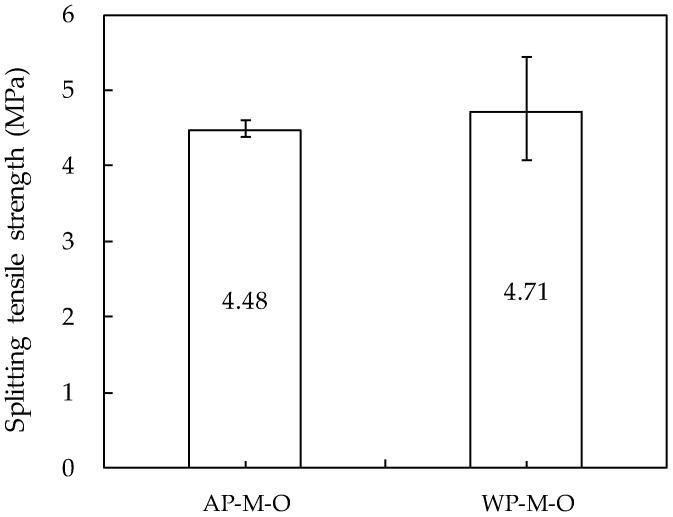
Splitting tensile strength.

**Table 1 materials-14-05888-t001:** Mixture proportioning.

W/B (%)	Unit Weight (kg/m^3^)				Admixture (%)	
	W	C	SF	S	VMA	HRWRA
38.4	250	586	66	1310	2.0	0.8

Note: W: water; C: cement; B: binder; SF: silica fume; S: sand; VMA: viscosity-modifying agent; HRWRA: high-range water-reducing agent.

**Table 2 materials-14-05888-t002:** Classification of specimens according to specimen manufacturing method and test method.

	Parts	Compressive Strength	Flexural Tensile Strength	Interlayer Bond Strength	Splitting Tensile Strength
Direct casting	-	AP-M-O	-	-	AP-M
		AP-M-X	-	-	-
		WP-M-O	-	-	WP-M
		WP-M-X	-	-	-
Extracting from parts	AP-4La	AP-CO	-	-	-
	AP-2La	AP-CU	AP-CU	AP-CU	-
	WP-4La	WP-CO	-	-	-
	WP-2La	WP-CU	WP-CU	WP-CU	-
	WP-2La-15	WP-CU-15	WP-CU-15	WP-CU-15	-

Note: AP: printed in air; WP: printed underwater; M: direct casting in cylinder molds; -O: compaction by tamping rod; -X: no compaction by tamping rod; 4La: parts additively manufactured in 4 layers; 2La: parts additively manufactured in 2 layers; 2La-15: parts additively manufactured in 2 layers with an interlayer time gap of 15 min; CO: coring parts; CU: cutting parts.

**Table 3 materials-14-05888-t003:** Result of property evaluation according to whether specimens were compacted by a compaction rod or not.

Specimen	Density at 7 Days, kg/m^3^	Compressive Strengthat 7 Days, MPa	Elastic Modulus at 7 Days, GPa
AP-M-O	2091 (13.6) [98.6%]	49.6 (2.37) [93.5%]	23.2 (0.81) [97.3%]
AP-M-X	2122 (5.5)	53.1 (0.19)	23.8 (0.30)
WP-M-O	2092 (13.6) [98.7%] <100.0%>	49.8 (1.33) [99.0%] <100.5%>	24.2 (0.54) [103.1%] <104.4%>
WP-M-X	2119 (9.5) <99.9%>	50.3 (1.31) <94.8%>	23.5 (1.04) <98.5%>

( ): standard deviation; [ ]: rate compared to specimens without tamping rod compaction; < >: rate compared to specimens printed in air.

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
