# Peer review of "Comparison of Properties of 3D-Printed Mortar in Air vs. Underwater"

_materials, 2021, doi:10.3390/ma14195888_

Round 1

Reviewer 1 Report

This work investigates the behavior of 3D-printed concrete in air vs. under-2 water. This work is within the scope of the journal.

Language is overall good.

The references in the study need to be enriched. 3D printed concrete is a popular subject in literature nowadays and findings from the last two years should be also considered and included in this work, highlighting the contribution to the field of this work, which is now missing and should be more clearly presented in the introduction section of the manuscript.

The concrete that can be solidified under is specific and is not necessarily able to be used in air, so the idea of comparing the same mixture, which is a fundamental principle in research needs to be further analyzed and discussed here, to enhance the value of the study results.

Line 70, "with no coarse aggregates" how this was determined?

Please change the term "printing" with "3D printing" were applicable throught the manuscript

How the parameters for the mixture and the experiment were selected, should be explained and appropriate references should be added, were needed.

The model of the concrete 3D printing machine and its specs should be described in the manuscript.

Were the specimens’ dimensions and specs according to a standard? If not, how were they selected and how much their shape and size affect the results of the experiment, should be discussed.

The model, the type and the specs of the machines used for the tensile, compression and flexural tests should be mentioned, along with the experimental setup for the tests. A picture from the actual tests should be included in the manuscript to depict the experimental setup.

For the flexural test it should be mentioned that it was a three-point-bending test

Figure 9, as mentioned in the text, is not taken by the authors, do the authors acquired the required license to use it in their work?

Most of the 3.1 section should be moved to the methodology section of the manuscript.

Figure 12 is confusing. Why are the stress-strain graphs not starting from 0,0? Also, are these graphs from the compression tests? This should be mentioned in the caption.

In figure 14 why the deviation is not symmetrical to the average value? Same with figure 15.

In figure 14 the deviation in the third and the sixth case are very high, why this is happening? Also, such results are unreliable and should be considered in the study.

The manuscript reads like a technical report with no actual discussion about the results, why this outcome is expected or not, no evaluation with literature and no analysis or even comparison between the results determined in the experiments. Results should be evaluated and discussed in the manuscript.

Reviewer 2 Report

The manuscript entitled: “Comparison of properties of 3D-printed mortar in air vs. underwater” is relevant for the Materials journal. It based on original laboratory research. The topic of the manuscript is up-to-date and important for future perspectives of additive technologies. The analysis is supported by proper literature. The manuscript is well organized, but requires some changes / clarifications such as:

  • Introduction: line 60, please consider additional literature: https://doi.org/10.1016/j.conbuildmat.2020.121649 and https://doi.org/10.3390/jmse8010026
  • Materials and Methods: line 90 – add more information about VMA (producer / supplier).
  • Materials and Methods: line 94 – add more information about HRWRA (producer / supplier).
  • Materials and Methods: point 2.2. add more information about used equipment, including type model, it was bought form the market or own production; what is the size of the equipment and working area; what is the size of water-tank etc.
  • Materials and Methods: point 2.2. add more information about used software or other type of management of the 3D printing process (automated or manual?).
  • Figure 4 – add information about the depth (in the text related to this figure).
  • Materials and Methods: point 2.4. what was the number of specimens for the each test?
  • Figure 8 please explain why compressive strength test was not made on cubic or cylindrical specimen (figure 8c). What was the potential influence of the other shape of samples? Add explanation in the text.
  • Figure 9 requires additional description what is exactly presented.
  • Materials and Methods: please add information about the water temperature and condition test “on the air” (temperature and humidity).
  • Results and discussion: clarify the statements in line 308. Normally the pressure in the water is higher and it increased with depth.
  • Please add more detailed discussion in point 3.5 and 3.7.

Round 2

Reviewer 1 Report

The revised version of the manuscript is significantly improved in its technical aspects. Most of the comments of this reviewer have been adequately replied and corresponding amendments have been made in the revised version of the manuscript. The only comment that the requires additional changes in the manuscript is :

“- The concrete that can be solidified under is specific and is not necessarily able to be used in air, so the idea of comparing the same mixture, which is a fundamental principle in research needs to be further analyzed and discussed here, to enhance the value of the study results.”

Authors replied to this comment, but also corresponding changes and comments must be added in the manuscript, according to their reply.

So, manuscript can be published after this minor change.

Author Response

- The revised version of the manuscript is significantly improved in its technical aspects. Most of the comments of this reviewer have been adequately replied and corresponding amendments have been made in the revised version of the manuscript.

Answer: We would like to thank you for your excellent comments which significantly improved the quality of our paper.

- The only comment that the requires additional changes in the manuscript is :

“- The concrete that can be solidified under is specific and is not necessarily able to be used in air, so the idea of comparing the same mixture, which is a fundamental principle in research needs to be further analyzed and discussed here, to enhance the value of the study results.”

Authors replied to this comment, but also corresponding changes and comments must be added in the manuscript, according to their reply.

Answer: As you commented, the corresponding comment is added to the 4. Conclusion, as follows.

“These experimental results were analyzed within the possible range based on existing theories and previous researches. Some points that have not yet been clearly identified will be analyzed in-depth through additional research.”

Reviewer 2 Report

The article has been significantly improved. Ir requires only slight text editing.

Author Response

We would like to thank you for your excellent comments which significantly improved the quality of our paper. Authors will carefully perform text editing before publication.